Altered expression profile of glycolytic enzymes during testicular ischemia reperfusion injury is associated with the p53/TIGAR pathway: effect of fructose 1,6-diphosphate

Al-Maghrebi May malmaghrebi@hsc.edu.kw 1
Renno Waleed M. 2
1 Faculty of Medicine—Department of Biochemistry, Kuwait University , Jabriyah , Kuwait
2 Faculty of Medicine—Department of Anatomy, Kuwait University , Jabriyah , Kuwait
Sterneck Esta
Electronic publication date: 2016 Jul 5
Publication date: 2016
Volume: 4
Electronic Location ID: e2195
Received 2016 Feb 28; Accepted 2016 Jun 8
Copyright: ©2016 Al-Maghrebi and Renno
Copyright year: 2016
Copyright holder: Al-Maghrebi and Renno
License: This is an open access article distributed under the terms of the Creative Commons Attribution License, which permits unrestricted use, distribution, reproduction and adaptation in any medium and for any purpose provided that it is properly attributed. For attribution, the original author(s), title, publication source (PeerJ) and either DOI or URL of the article must be cited.
License URL: https://creativecommons.org/licenses/by/4.0/

Keywords: Testicular ischemia reperfusion injury; Gene expression regulation; Apoptosis; Oxidative stress; Fructose 1,6-diphosphate; Glycolytic enzymes

Funding: Kuwait University Research Sector MB 02/15 This study was funded by Kuwait University Research Sector grant MB 02/15. The funders had no role in study design, data collection and analysis, decision to publish, or preparation of the manuscript.

==============================
Background. Testicular ischemia reperfusion injury (tIRI) is considered the mechanism underlying the pathology of testicular torsion and detorsion. Left untreated, tIRI can induce testis dysfunction, damage to spermatogenesis and possible infertility. In this study, we aimed to assess the activities and expression of glycolytic enzymes (GEs) in the testis and their possible modulation during tIRI. The effect of fructose 1,6-diphosphate (FDP), a glycolytic intermediate, on tIRI was also investigated.

Methods. Male Sprague-Dawley rats were divided into three groups: sham, unilateral tIRI, and tIRI + FDP (2 mg/kg). tIRI was induced by occlusion of the testicular artery for 1 h followed by 4 h of reperfusion. FDP was injected peritoneally 30 min prior to reperfusion. Histological and biochemical analyses were used to assess damage to spermatogenesis, activities of major GEs, and energy and oxidative stress markers. The relative mRNA expression of GEs was evaluated by real-time PCR. ELISA and immunohistochemistry were used to evaluate the expression of p53 and TP53-induced glycolysis and apoptosis regulator (TIGAR).

Results. Histological analysis revealed tIRI-induced spermatogenic damage as represented by a significant decrease in the Johnsen biopsy score. In addition, tIRI reduced the activities of hexokinase 1, phosphofructokinase-1, glyceraldehyde 3-phosphate dehydrogenase, and lactate dehydrogenase C. However, mRNA expression downregulation was detected only for hexokinase 1, phosphoglycerate kinase 2, and lactate dehydrogenase C. ATP and NADPH depletion was also induced by tIRI and was accompanied by an increased Malondialdehyde concentration, reduced glutathione level, and reduced superoxide dismutase and catalase enzyme activities. The immunoexpression of p53 and TIGAR was markedly increased after tIRI. The above tIRI-induced alterations were attenuated by FDP treatment.

Discussion. Our findings indicate that tIRI-induced spermatogenic damage is associated with dysregulation of GE activity and gene expression, which were associated with activation of the TIGAR/p53 pathway. FDP treatment had a beneficial effect on alleviating the damaging effects of tIRI. This study further emphasizes the importance of metabolic regulation for proper spermatogenesis.

Introduction

Testicular ischemia reperfusion injury (tIRI) represents the events that occur during testicular torsion and detorsion (TDD) (Filho et al., 2004). Prompt surgical intervention to counter rotation of the spermatic cord is necessary to relieve the acute ischemic episode. However, it has been established that tIRI induces the generation of reactive oxygen species (ROS) that trigger an array of signaling molecules, causing progressive damage to the structure and function of the testis (Filho et al., 2004; Antonuccio et al., 2006; Minutoli et al., 2011). A potentially negative outcome of tIRI is the increased risk of impaired spermatogenesis and reduced male fertility. Throughout spermatogenesis, germ cells have unique metabolic needs and utilize different metabolic pathways for energy production to ensure proper development (Boussouar & Benahmed, 2004; Rato et al., 2012). Therefore, the maintenance of metabolic requirements necessitates the cooperation of germ cells with Sertoli cells (SCs) to ensure proper spermatogenesis (Alves et al., 2014). Spermatogenesis is under hormonal control that involves SCs; thus, it can be inferred that the glycolytic metabolism of SCs might be affected by this hormonal signaling (Alves et al., 2013). Moreover, the metabolic status of the reproductive system is vital to maintain germ cell energy demands and survival. Several studies have reported on the implications of metabolic defects on male infertility in patients with Klinefelter syndrome, diabetes mellitus, or obesity (Alves et al., 2016a; Alves et al., 2015; Martins et al., 2015; Rato et al., 2015).

During spermatogenesis, germ cells interchangeably utilize lactate and glucose as energy sources. Spermatids predominantly use the TCA cycle for energy production, whereas spermatogonia depend on glycolysis for energy production (Robinson & Fritz, 1981; Riera et al., 2002). SCs act as lactate producers in the testis by sustaining a high glycolytic flux, thus exhibiting Warburg-like metabolic behavior (Oliveira et al., 2015). This is very important for supporting the energetic needs of developing germ cells. Most of the glucose is converted to lactate, which is then converted to pyruvate by the enzyme lactate dehydrogenase C (LDHC). Interestingly, approximately 25% of the produced pyruvate becomes oxidized during the TCA cycle. This reaction is coupled with NADPH production, an indicator of the pentose phosphate pathway (PPP). Surplus glucose undergoes glycolysis to produce pyruvate that enters the mitochondria and becomes oxidized and decarboxylated by pyruvate dehydrogenase to form acetyl CoA that enters the TCA cycle. During this process, ATP is formed by ADP phosphorylation via the electron transport chain (Rato et al., 2012). Several glycolytic enzymes (GEs), including hexokinase isoenzyme 1 (HK1S), glucose-6-phosphate isomerase (GPI), phosphoglycerate kinase (PGK2), sperm-specific glyceraldehyde 3-phosphate (GAPDHS), and pyruvate kinase (PKS), have been identified and reported to play a key role in regulating glycolysis in germ cells (Gupta, 2013). An interesting glycolysis intermediate, fructose 1,6-diphosphate (FDP), has been shown to exert protective effects in several models of ischemia and hypoxia (Didlake et al., 1989; Farias, Smith & Markov, 1990; Karaca et al., 2002). It was also suggested that FDP can activate the PPP for anaerobic ATP production (Kelleher et al., 1995; Espanol et al., 1999). The PPP is involved in the regulation of cellular redox via NADPH supplementation. NADPH is considered a regulator of cellular redox potential by preserving cellular levels of glutathione and catalase (Ben-Yosef, Boxer & Ross, 1996; Salvemini et al., 1999). Similarly, FDP was found to maintain cellular levels of glutathione and catalase in addition to upregulating the activity of glucose-6-phosphate dehydrogenase, the rate-limiting enzyme of PPP. Therefore, it is highly likely that FDP suppresses oxidative stress through PPP activation (Ahn et al., 2002).

In the present study, we evaluated the mRNA expression and activity of the major GEs required for energy production in the testis during tIRI. In parallel, testicular oxidative status and germ cell apoptosis were also assessed. Due to its reported protective and antioxidant effects, we investigated the effects of the exogenous administration of FDP on spermatogenesis, expression of GEs, and energy production in the testis. The effects of tIRI-induced ROS on the expression of p53 and its downstream transcriptional target TP53-induced glycolysis and apoptosis regulator (TIGAR) were also studied.

Materials and Methods

Ethics statement

The animal experimental protocol and procedures used in this study complied with the guidelines of the ethics committee on animal research at Kuwait University. The ethical use of animals at Kuwait University is in accordance with the guidelines of the International Council for Laboratory Animal Sciences (ICLAS). Male adult Sprague-Dawley rats (Charles River, Waltham, MA, USA) were acclimated to standard laboratory conditions of a 12-h light/12-h dark cycle at 25 °C, and were fed a standard diet and tap water ad libitum.

Surgical procedure

The rats (n = 18, 200–250 g, 8 weeks old) were divided randomly into three groups of six rats each. The three groups were: sham, tIRI, and tIRI + FDP. The surgical procedure has been described previously (Al-Maghrebi, Renno & Al-Ajmi, 2012). Briefly, all rats were anesthetized with 50 mg/kg ketamine (Tekam, Hikma Pharmaceuticals, Amman, Jordan) and 2 mg/kg xylazine (Rompun, Bayer GmbH, Leverkusen, Germany). The incision area was clean-shaven and disinfected with betadine. Sham rats underwent a standard ilioinguinal incision at the left side, and the left testis was exposed for 60 min before placing it again into the scrotal sac followed by incision suturing. Sham animals were sacrificed after 4 h. The rats subjected to tIRI underwent a unilateral ischemic injury by occluding the left testicular artery with a non-traumatic microvascular clamp (700 g of pressure) (Cat. No. RS-7440; Roboz Surgical Instruments Co., Gaithersburg, MD, USA) to cut off the blood supply to the testes for 1 h. Thirty minutes prior to testis reperfusion, the rats received an intraperitoneal injection (i.p.) injection of 300 µl of saline (vehicle). Blood flow was resumed after 1 h of ischemia by clamp removal, and testis reperfusion was allowed for 4 h before animal sacrifice. A similar procedure was followed with the third group that underwent tIRI + FDP, in which saline was substituted with a dose of 2 g/kg FDP. FDP (Cat. No. F6803; Sigma-Aldrich, St. Louis, MO, USA) was administered as an i.p. of 2 mg/kg 30 min prior to reperfusion. The selected dose and method of delivery were based on prior studies (Zhou et al., 2014; Planas et al., 1993). Zhou and colleagues (2014), showed that an i.p. FDP dose of 500 or 1,000 mg/kg provided neuroprotection in immature rats suffering from repeated febrile convulsions. Palanas and colleagues (1993), demonstrated that in contrast to lower i.p. doses of FDP (0.5 or 1 g/kg) or the orally administered dose of 0.5 g/kg, an i.p. injection of 2 g/kg FDP had the highest protective action (80%) within 1 h of administration and persisted for 5 h. Furthermore, albino Swiss mice showed no toxicity symptoms after an i.p. administration of 800 mg/kg FDP. The availability of FDP for 5 h and low toxicity are important to evaluate its protective effects in our experimental design. For all three animal groups, the right contralateral testes were used as a positive internal control.

Histological examination

The harvested testes were immediately immersed in Bouin’s fixative for 24 h, washed with PBS, and embedded in paraffin. Hematoxylin and eosin (H&E) staining was used to stain 4-µm tissue sections. Spermatogenesis was evaluated by measuring the tissue biopsy score (TBS) using the Johnson scoring system, which is based on rating germ cell maturation in each seminiferous tubule using a score of 1–10 (Johnsen, 1970). In a blinded manner, four slides from each testis (six contralateral and six ipsilateral testes) were used for scoring.

Detection of apoptosis

Dewaxed and rehydrated 4-µm tissue sections were treated with proteinase K followed by incubation with the TUNEL reaction mixture at 37 °C and then were mounted with DAPI. Staining of non-apoptotic free DNA 3′ ends was eliminated by adjusting the manufacturer’s protocol (Cat. No. 11684795910; Roche-Diagnostics, Mannheim, Germany). TUNEL-stained nuclei were analyzed using the LSM 700 confocal laser scanning microscope (Carl Zeiss Micro-Imaging, München, Germany). TUNEL-stained nuclei were scored using 100 random seminiferous tubules from two slides/testis/rat. Images were acquired at 400× and 100× magnification for counting and presentation purposes, respectively. The investigator was blinded to the experimental group identity during the scoring process, and the data are presented as the mean ± SD.

Immunohistochemistry

Each testis paraffin block was cut into 4-mm sections on silane-coated slides. The avidin-biotin complex method was used for p53 and TIGAR immunohistochemical staining. Tissues were rehydrated in a graded alcohol series followed by microwave retrieval using citrate buffer (0.01 M, pH 6). The tissues were then treated for 30 min in 3% H2O2, washed in PBS and blocked in blocking solution (Cat. No. 85-8943; Invitrogen, Frederick, MD, USA). The processed slides were incubated overnight with primary antibodies (1:100 dilution) for p53 (Cat. No. (DO-1) sc-126; Santa Cruz Biotechnology, Santa Cruz, CA, USA) and for TIGAR (Cat. No. (Y-20) sc-68239; Santa Cruz Biotechnology, Santa Cruz, CA, USA). The slides were then washed and treated with a Histostain-Plus IHC Kit, HRP, and broad-spectrum secondary antibody (60:40) (Cat. No. 85-8943; Thermo Fisher Scientific, Waltham, MA, USA) for 30 min at room temperature. The slides were then washed three times in PBS and treated with HRP-Streptavidin (Cat. No. N200; Thermo Fisher Scientific, Waltham, MA, USA) for 30 min at room temperature. Color development was then carried out using the Impact DAB kit (Cat. No. SK-4105; Vector Labs, Burlingame, CA, USA) for 30 s or until the desired brown color was obtained as seen under the microscope. The slides were then washed in distilled water and counterstained with hematoxylin for 5 min, followed by bluing under tap water for 5 min. The slides were dehydrated through a graded series of alcohol from 50% to absolute alcohol and then were cleared in xylene. Finally, a coverslip was mounted on top of the tissue sections using histology DPX mountant. The mean immunolabeling concentration was assessed by measuring the color intensity (Sum (Area) (pixel2) using Cell Sens Dimension Software (Olympus DP 71 camera) in the three experimental groups. Slide analysis was performed in a blinded manner.

RT and Real-time PCR

Total RNA was purified from frozen testicular tissue samples using TRIzol (Invitrogen, USA) following the manufacturer’s instructions. Total RNA was reverse transcribed into complementary DNA (cDNA) using a high-capacity cDNA reverse transcription kit (RT) (Thermo Fisher Scientific, Waltham, MA, USA). The reaction mixtures included 20 U of reverse transcriptase, 10 µl of first-strand buffer, random primers, 0.5 mM dNTP mix and 10 mM DDT. RT reactions were carried out at 55 °C for 50 min. The gene expression levels of the following GEs were quantitated using real-time PCR: hexokinase (hk1, Cat. No. Rn00562436_m1), glucose-6-phosphate isomerase (gpi, Cat. No. Rn01475756_m1), 6-phosphofructokinase 1 (pfk, custom made), glyceraldehyde 3-phosphate dehydrogenase Sperm (gapdhs, Cat. No. Rn01476455_m1), phosphoglycerate kinase 2 (pgk2, Cat. No. Rn01511987_s1), and lactate dehydrogenase C (ldhc, Cat. No. Rn00568562_m1). The mRNA levels for the p53 upregulated modulator of apoptosis (puma, Cat. No. Rn00597992_m1) and survivin (birc5, Cat. No. Rn00574012_m1) were also measured. Gene-specific Taqman assays were mixed with TaqMan®universal PCR master mix (Thermo Fisher Scientific, Waltham, MA, USA) in a 96-well plate. The reaction conditions recommended by the manufacturer were used in compliance with the ABI Prism 7000 SD system (Thermo Fisher Scientific, Waltham, MA, USA). The relative mRNA expression was calculated using the 2−ΔΔCT method (Livak & Schmittgen, 2001). A rat β-actin (actb, Cat. No. Rn00667869_m1)-specific Taqman assay was used as an endogenous control.

Protein purification

Total protein crude extracts were prepared from harvested testicular tissue. The tissues were homogenized in a lysis buffer (20 mM Tris, 150 mM NaCl, 10 mM Na2EDTA, 10 mM EGTA, 1% Triton X-100, 1 mM Na3VO4, 25 mM NaF, 1 µg/ml leupeptin, and 1 mM PMSF) using a tissue homogenizer. The protein concentrations were measured using the Bradford assay (Bio-Rad, Hercules, CA, USA) and bovine serum albumin as the standard.

Biochemical assays

Colorimetric assays were purchased from Sigma-Aldrich (St. Louis, MO, USA) to measure the enzymatic activities of HK1 (Cat. No. MAK091), GPI (Cat. No. MAK103), PFK1 (Cat. No. MAK093), GAPDH (Cat. No. MAK277), PGK2 (Cat. No. P7634), and LDHC (Cat. No. MAK066) following the manufacturer’s protocols. The protein expression of total p53 and p-p53 (ser 15) was measured using an ELISA kit (Cat. No. PEL-P53-S15-T-1; RayBiotech, Inc., Norcross, GA, USA) following the manufacturer’s protocol. The levels of ATP (Cat. No. MAK190) and NADPH (Cat. No. MAK038) were measured using their respective colorimetric assays according to the manufacturer’s recommendations (Sigma-Aldrich, St. Louis, MO, USA). The testicular levels of glutathione (GSH, Cat. No. CS0260), malondialdehyde (MDA, Cat. No. MAK085) and the antioxidant enzymes superoxide dismutase (SOD, Cat. No. 19160) and catalase (CAT, Cat. No. CAT100) were determined using their respective colorimetric assays (Sigma-Aldrich, St. Louis, MO, USA) following the manufacturer’s protocol.

Statistical analysis

Statistical analysis for the obtained data was performed using GraphPad Prism (v6.0). All of the data are presented as the mean ± standard deviation (SD). Grubb’s test and/or ROUT test were used to eliminate any outliers from the normal data distribution. Multiple group comparisons were performed using one-way analysis of variance (ANOVA) followed by the Holm-Sidak multiple comparisons test for the comparison of mean values. Statistical significance was accepted as p < 0.05.

Figure 1 Histological analysis of testicular tissue.

H&E sections of rat ipsilateral testes showing low (A, B and C; 10×) and high (D, E and F; 40×) magnifications of the histological changes from the three experimental groups. (A and D) Representative images of ipsilateral testes from the sham group showing a normal seminiferous tubule structure and normal germ cell layer arrangement. (B and E) Histological sections of ipsilateral testes from the tIRI group showing seminiferous tubule atrophy and disrupted germ cell layers. (C and F) Examples of histological findings of the FDP-treated group (2 mg/kg i.p.) showing preserved histological morphology of the seminiferous tubules.

Results

Testicular spermatogenesis

Figure 1 shows H&E testicular sections from the sham, tIRI, and FDP-treated groups. In contrast to the normal histological appearance of testicular tissue in the sham and FDP-treated groups, the ipsilateral testes in the tIRI group showed seminiferous tubular atrophy, disruption of germ cell layers, and spermatogenic arrest. Compared with sham, the mean TBS in the ipsilateral testes from the tIRI group was significantly lower (9.33 ± 1.03 vs. 5.67 ± 0.52, p < 0.0001), which was normalized after FDP treatment (7.67 ± 0.52, vs. 5.67 ± 0.52, p = 0.0002). The contralateral testes revealed no significant differences in testis histology or TBS among the three experimental groups.

mRNA expression of glycolytic enzymes

The relative mRNA expression was obtained and calculated for all of the evaluated genes (Table 1). The mRNA expression for gpi and pgk2 showed no significant changes among the three experimental groups. However, significant downregulation of the mRNA expression of hk1 (0.76 ± 0.46 vs. 1.00 ± 0.26, p < 0.05), pfk (0.66 ± 0.39 vs. 1.00 ± 0.27, p < 0.05), gapdhs (0.73 ± 0.39 vs. 1.00 ± 0.32, p < 0.05), and ldhc (0.76 ± 0.46 vs. 1.00 ± 0.26, p < 0.05) was calculated in the tIRI group compared with sham levels and was restored to almost sham levels in the FDP-treated group. The contralateral testes revealed no significant differences in mRNA expression among the three experimental groups.

Table 1 Relative mRNA expression [RQ (2−ΔΔCT)] of glycolytic enzymes.

Rats received an i.p. injection of FDP (2 mg/kg) 30 min prior to reperfusion. Data analysis was determined by the one way analysis of variance (ANOVA) accompanied by the Holms-Sidak multiple comparisons test. Data are presented as mean ± SD (n = 6).

Gene name		RQ (2−ΔΔCT)	
		
		Sham	tIRI	FDP	
Hk1	I	1.00 ± 0.26	0.76 ± 0.46a	1.06 ± 0.49b	
	C	1.00 ± 0.09	1.05 ± 0.33	1.13 ± 0.26	
Gpi	I	1.00 ± 0.31	0.91 ± 0.64	0.93 ± 0.45	
	C	1.00 ± 0.22	1.13 ± 0.33	1.00 ± 0.54	
Pfk1	I	1.00 ± 0.27	0.66 ± 0.39a	0.91 ± 0.21b	
	C	1.00 ± 0.39	1.10 ± 0.59	1.19 ± 0.28	
Gapdhs	I	1.00 ± 0.32	0.73 ± 0.39a	0.92 ± 0.30b	
	C	1.00 ± 0.07	1.02 ± 0.09	1.05 ± 0.10	
Pgk2	I	1.00 ± 0.24	1.05 ± 0.55	1.19 ± 0.35	
	C	1.00 ± 0.22	0.92 ± 0.24	1.09 ± 0.50	
Ldhc	I	1.00 ± 0.26	0.76 ± 0.46a	1.06 ± 0.46b	
	C	1.00 ± 0.09	1.05 ± 0.33	1.14 ± 0.26	
Actin, beta	I	–	–	–	
	C	–	–	–	
Notes.

a tIRI compared to sham.

b FDP compared to tIRI.

I Ipsilateral

C Contralateral

Glycolytic enzyme activities

The enzymatic activities of HK1 (1.11 ± 0.27 vs. 2.08 ± 0.30, p = 0.0034), PFK1 (0.78 ± 0.10 vs. 1.33 ± 0.24, p = 0.0016), GAPDHS (1.05 ± 0.21 vs. 1.49 ± 0.14, p = 0.0016) and LDHC (1.07 ± 0.17 vs. 2.02 ± 0.41, p < 0.0001) were significantly decreased as a result of tIRI compared with sham levels (Table 2). These activities were significantly increased in the FDP-treated group compared with that in the tIRI group (HK1: 1.87 ± 0.38, p = 0.0260; PFK1: 1.30 ± 0.17, p = 0.0029; GAPDHS: 1.46 ± 0.28, p = 0.0040; LDHC: 1.78 ± 0.19, p = 0.0017). Although there were notable decreases in the activities of GPI and PGK2 in the tIRI group, the decreases were not significantly different from those of the sham or FDP-treated group. The contralateral testes revealed no significant difference in enzymatic activities among the three experimental groups.

Table 2 Activities of glycolytic enzymes.

Rats received an i.p. injection of FDP (2 mg/kg) 30 min prior to reperfusion. Enzyme activities are expressed in milliunits/µg protein and data are presented as mean ± SD (n = 6). Data analysis was determined by the one way analysis of variance (ANOVA) accompanied by the Holms-Sidak multiple comparisons test.

		Sham	tIRI	p valuea	FDP	p valueb	
HK1	I	2.08 ± 0.30	1.11 ± 0.27	0.0034	1.87 ± 0.38	0.0260	
	C	1.72 ± 0.43	1.71 ± 0.48	ns	1.55 ± 0.70	ns	
GPI	I	0.91 ± 0.11	0.77 ± 0.12	0.2046	0.71 ± 0.16	0.8686	
	C	0.73 ± 0.14	0.88 ± 0.10	ns	0.79 ± 0.09	ns	
PFK1	I	1.33 ± 0.24	0.78 ± 0.10	0.016	1.30 ± 0.17	0.0029	
	C	1.2 ± 0.21	1.08 ± 0.30	ns	1.42 ± 0.25	ns	
GAPDHS	I	1.49 ± 0.14	1.05 ± 0.21	0.0016	1.46 ± 0.28	0.0040	
	C	1.36 ± 0.15	1.32 ± 0.18	ns	1.37 ± 0.16	ns	
PGK2	I	0.98 ± 0.21	0.77 ± 0.09	0.1388	0.81 ± 0.14	0.9894	
	C	0.92 ± 0.19	0.84 ± 0.15	ns	0.83 ± 0.17	ns	
LDHC	I	2.02 ± 0.41	1.07 ± 0.17	<0.0001	1.78 ± 0.19	0.0017	
	C	2.02 ± 0.34	2.02 ± 0.16	ns	1.84 ± 0.43	ns	
Notes.

a tIRI compared to sham.

b FDP compared to tIRI.

I Ipsilateral

C Contralateral

Testicular levels of ATP and NADPH

The levels of ATP and NADPH were evaluated in the three experimental groups (Fig. 2). In the tIRI group, testicular tissue showed significantly reduced ATP levels (0.41 ± 0.08 vs. 0.59 ± 0.05, p = 0.0087) and NADPH levels (1.75 ± 0.36 vs. 2.88 ± 0.30, p = 0.0004) compared with those in the sham group. FDP-treated rats exhibited sham-like levels of ATP (0.57 ± 0.12 vs. 0.41 ± 0.08, p = 0.0286) and NADPH (2.53 ± 0.32 vs. 1.75 ± 0.36, p = 0.0168) compared with tIRI. The contralateral testes revealed no significant differences in the evaluated parameters among the three experimental groups.

Figure 2 FDP effects on ATP and NADPH levels.

(A) tIRI-induced ATP depletion (∗p = 0.0087) was preserved in the FDP-treated group (#p = 0.0286). (B) The tIRI-induced decrease in NADPH levels (∗p = 0.0004) was alleviated by FDP treatment (#p = 0.0168). Data analysis was performed using one-way analysis of variance (ANOVA) followed by the Holm-Sidak multiple comparisons test. Data are presented as the mean ± SD (n = 6). The significance of the data is indicated with ∗ compared with the sham group and # compared with the tIRI group.

Testicular oxidative stress

In the tIRI group, significantly low GSH (0.33 ± 0.03 vs. 0.25 ± 0.04, p = 0.0002) and high MDA levels (1.68 ± 0.35 vs. 1.05 ± 0.20, p < 0.0001) were accompanied by decreased activities of CAT (82.55 ± 8.52 vs. 95.32 ± 3.91, p = 0.0002) and SOD (81.23 ± 0.95 vs. 95.67 ± 3.59, p < 0.0001) compared with sham (Table 3). FDP treatment reduced testicular oxidative stress as indicated by normalized levels of GSH (0.24 ± 0.03, p = 0.0002), MDA (1.04 ± 0.25, p < 0.0001), and CAT (94.75 ± 4.10, p = 0.0003) and SOD (92.62 ± 1.87, p = 0.0004) activities. The contralateral testes showed no significant changes in these parameters.

Table 3 Levels of oxidative stress markers.

Rats received an i.p. injection of FDP (2 mg/kg) 30 min prior to reperfusion. Parameter units: GSH (nmol/µg); MDA (µM); SOD (inhibition rate %); and CAT (units/ml). Data analysis was determined by the one way analysis of variance (ANOVA) accompanied by the Holms-Sidak multiple comparisons test. Data are presented as mean ± SD (n = 6).

		Sham	tIRI	p valuea	FDP	p valueb	
GSH	I	0.25 ± 0.04	0.33 ± 0.03	0.0002	0.23 ± 0.01	<0.0001	
	C	0.22 ± 0.02	0.21 ± 0.02	ns	0.19 ± 0.01	ns	
MDA	I	1.05 ± 0.20	1.68 ± 0.35	<0.0001	1.04 ± 0.25	<0.0001	
	C	1.03 ± 0.07	0.98 ± 0.06	ns	1.01 ± 0.08	ns	
SOD	I	95.67 ± 3.59	81.23 ± 0.95	<0.0001	92.62 ± 1.87	0.0004	
	C	94.12 ± 2.48	92.03 ± 3.52	ns	90.53 ± 7.23	ns	
CAT	I	95.32 ± 3.91	82.55 ± 8.52	0.0002	94.75 ± 4.10	0.0003	
	C	94.07 ± 4.17	92.32 ± 3.19	ns	97.09 ± 0.98	ns	
Notes.

a tIRI compared to sham.

b FDP compared to tIRI.

I Ipsilateral

C Contralateral

Figure 3 Germ cell apoptosis (GCA) assessed by TUNEL immunohistofluorescence.

(A, D and G) Representative images of GCA in sham rats. (B, E and H) Increased number of TUNEL-positive nuclei in the tIRI group. (C, F and I) FDP-treated group showing a diminished number of TUNEL-positive nuclei. Fluorescence staining used in the images: A, D, and G = TUNEL; B, E, and H = DAPI, and C, F, and I = Merged.

Figure 4 Phosphorylation of p53 and modulation of its downstream genes.

(A) Increased p53 phosphorylation (ser 15) was induced by tIRI (∗p < 0.0001) and was inhibited by FDP treatment (#p < 0.0001). Real-time PCR was used to measure the relative mRNA expression of (B) PUMA and (C) survivin. PUMA mRNA levels were increased during tIRI (∗p < 0.05), whereas survivin mRNA was decreased (∗p < 0.05). FDP treatment normalized the relative mRNA expression of PUMA (#p < 0.05) and survivin (#p < 0.05). Data analysis was performed using one-way analysis of variance (ANOVA) followed by the Holm-Sidak multiple comparisons test. Data are presented as the mean ± SD (n = 6). The significance of the data is indicated with ∗ compared with the sham group and # compared with the tIRI group.

Figure 5 Immunoexpression of p53 and TIGAR in paraffin-embedded ipsilateral testicular tissue.

(A and D) The sham group showing very weak immunostaining of the p53 and TIGAR antibodies, respectively, in the seminiferous tubules layers of the rat testes. (B and E) Examples of p53 and TIGAR immunostaining showing a notable increase in their immunoexpression after tIRI. (C and F) FDP treatment of tIRI-subjected rats showed a decrease in the p53 and TIGAR immunolabeling intensity in the testes compared with the tIRI group. Magnification = 40×.

Germ cell apoptosis and p53/TIGAR signaling

As shown in Fig. 3, ROS-induced DNA damage is clearly visible in the ipsilateral testes subjected to tIRI compared with the sham group (38.67 ± 8.98 vs. 0.83 ± 0.75, p < 0.0001); this damage was prevented by FDP treatment (9.83 ± 4.62 vs. 38.67 ± 8.98, p < 0.0001). As shown in Fig. 4, DNA damage was accompanied by increased p53 phosphorylation (ser 15) (1.47 ± 0.25 vs. 0.90 ± 0.11, p < 0.0001), upregulated PUMA mRNA expression f (1.71 ± 0.28 vs. 1.00 ± 0.06, p < 0.05), and downregulated survivin mRNA expression (0.66 ± 0.15 vs. 1.00 ± 0.16, p < 0.05) in the tIRI group compared with the sham group. Phosphorylation of p53 and the mRNA expression of PUMA and survivin reverted to sham levels in the FDP-treated group (0.96 ± 0.21 (p < 0.0001), 1.10 ± 0.15 (p < 0.05), and 1.14 ± 0.14 (p < 0.05), respectively). This result was also associated with significant increases in the immunostaining of both p53 and TIGAR in tIRI-subjected testes compared with the sham group (p53: 44,511 ± 14,731 vs. 1,053 ± 337, p < 0.0001; TIGAR: 223,212 ± 61,975 vs. 42,743 ± 22,900, p < 0.0001) (Fig. 5). FDP treatment significantly reduced the immunostaining of both proteins (p53: 5,757 ± 3,223 vs. 44,511 ± 14,731, p < 0.0001; TIGAR: 95,124 ± 48,736 vs. 223,212 ± 61,975, p < 0.0001).

Discussion

Testicular IRI is characterized by reduced spermatogenesis, which could affect testicular function in supporting male fertility. The regulation of carbohydrate metabolism and energy production is important in testicular cells to maintain proper spermatogenesis (Rato et al., 2012). Such processes are carried out through unique metabolic cooperation between somatic SCs and developing germ cells (Alves et al., 2014). Here, we tested the hypothesis that tIRI would affect the metabolic profile of the testis.

Glycolysis can be divided into two distinct stages: an energy-consuming stage to produce FDP from glucose, and a second stage comprising energy production via the degradation of FDP to pyruvate (Berg, Tymoczko & Stryer, 2002). Although SCs and germ cells express all of the enzymes of the glycolytic pathway, germ cells predominantly use the lactate produced by SCs for energy production (Oliveira et al., 2015; Boussouar & Benahmed, 2004). Our data indicate that the transcriptional deregulation of key regulatory GEs from the two glycolytic stages was coupled with respective reduced enzymatic activities, and both were associated with impaired spermatogenesis, increased oxidative stress and germ cell apoptosis. For germ cells, glycolysis is the major source of ATP required for sperm motility, capacitation, and fertilization (Miki, 2007; Mukai & Okuno, 2004; Travis et al., 2004; Williams & Ford, 2001). This result was supported by the localization of the major glycolytic enzymes HK1S, PFK, GAPDHS and LDHC as a unique cluster in the fibrous sheath of the flagellum and in close proximity to the dynein ATPase activity that generates the flagellar beat (Krisfalusi et al., 2006; Nakamura, Mori & Eddy, 2010; Tanii et al., 2007). The importance of their role in supporting sperm bioenergetics and function was further realized in knockout mice for GAPDHS−∕− (Miki et al., 2004) and PGK2−∕− (Danshina et al., 2010), which showed severely impaired fertility with glycolysis inhibition. Although spermatogenesis appeared normal in LDHC−∕− mice (Odet et al., 2013), the sperm fertilization capacity was greatly compromised due to the inability to metabolize lactate for energy production. In addition, immotile sperms had reduced PFK activity compared with motile sperms (Kamp et al., 2007). Furthermore, metal-treated human ejaculates were found to exhibit dose- and time-dependent effects on sperm motility due to the inhibition of GEs such as glucose-6-phosphatase, fructose 1,6-diphosphatase, glucose-6-phosphate isomerase, and lactic dehydrogenase (Kanwar et al., 1988). It was also suggested that the depletion of lactate levels in the testicular tissues of patients with Klinefelter syndrome, a common genetic cause of human infertility, was due to underlying alterations in the testicular metabolism supported by SCs (Alves et al., 2016a). Similarly, compromised levels of testicular lactate content were observed in type I diabetic men and were attributed to alterations in glycolysis-related transporters and enzymes and structural SC deformities, which might be related to impotency problems (Alves et al., 2015). Obesity is another metabolic disease that provides an immediate link between male infertility and deregulated metabolism (Alves et al., 2016b; Katib, 2015; Rato et al., 2014). Impaired sperm quality was associated with defects in energy metabolic pathways in rats fed a high-fat diet (Ferramosca et al., in press; Rato et al., 2013). Collectively, these data strongly suggest that the impairment of testicular glycolysis due to the deregulated expression and activity of GEs in testicular somatic or germ cells would compromise testicular function and could increase the risk of male infertility.

Regular and continuous oxygen supply to the testis is vital for its overall function and, most importantly, to promote proper spermatogenesis. Oxidative stress and a lowered antioxidant capacity are hallmarks of tIRI and are the bases of its pathophysiological consequences, among other factors (Turner & Lysiak, 2008; Al-Maghrebi, Kehinde & Anim, 2010). In our study, the decline in glycolytic flux during tIRI was accompanied by increased testicular oxidative stress as indicated by the low levels of NADPH, GSH, CAT and SOD. Under physiological conditions, it is thought that proliferating cells using aerobic glycolysis are naturally protected against OS due to the production of NADPH and GSH, which are natural antioxidant side products of the PPP (Kuehne et al., 2015). Thus, increased ROS production as a result of tIRI would disrupt the function of the enzymes and components of the metabolic and energy producing pathways in germ cells. A strong association was reported between increased ROS generation, abnormal semen parameters and male infertility (Ko, Sabanegh Jr & Agarwal, 2014). Recently, it was also reported that increased ROS levels in the serum and seminal fluid of infertile men negatively affected sperm mitochondrial respiration through the uncoupling of electron transport and ATP production (Ferramosca et al., 2013). Although considered non proliferative, SCs are characterized by a high glycolytic flux and are known as lactate producers due to their unique “Warburg-like metabolism” to support the continuous progression of spermatogenesis (Oliveira et al., 2015). This property indirectly renders SCs insensitive to physiological oxidative stress but not that induced by continuous external stimuli. Therefore, ROS can be considered a regulator of glycolytic flux in proliferative germ cells and non-proliferative SCs.

Oxygen supply insufficiency induced by ischemia or hypoxia also alters mitochondrial respiratory chain function, inhibits mitochondrial ATP synthase and subsequently reduces oxidative phosphorylation (Eales, Hollinshead & Tennant, 2016). Here, we demonstrated that testicular ATP levels are compromised in tIRI-subjected rats and were preserved after FDP treatment. Tissue reperfusion and the restoration of oxygen supply generate ROS, open mitochondrial pores and disrupt the proton balance, exacerbating ischemic injury and leading to ATP depletion (Baines, 2010; Halestrap, 2010). The beneficial effects of FDP in maintaining the intracellular ATP pool can occur at different levels. FDP is known to activate membrane Na+∕K+ ATPase by increasing intracellular Na+ and sustaining ionic equilibrium (Roig, Bartrons & Bermudez, 1997), to stimulate PFK activity (Farias, Willis & Gregory, 1986) and to act as an energy source by stimulating anaerobic ATP production (Bickler & Buck, 1996). In the hypothermic heart, FDP protection was related to the increased production of intracellular glycolytic energy (Hua et al., 2003). Physiologically, each FDP molecule can produce four ATP molecules instead of two ATPs produced by glucose under aerobic conditions by bypassing the first two ATP-consuming reactions. Therefore, it can be hypothesized that during tIRI, the availability of FDP prior to the onset of reperfusion could have rescued the cell from the decline in the HK and PFK activities that require ATP and acted as a glycolytic substrate for the next ATP-generating glycolytic reactions.

The role of p53 in regulating spermatogenesis is commonly accepted (Almon et al., 1993) because its knockout in mice leads to more undifferentiated spermatogonia compared with wild-type mice (Beumer et al., 1998). However, little is known about the p53 regulation of glycolytic flux in testicular cells. Here, we show that tIRI-induced p53 overexpression was associated with increased TIGAR expression, p53 phosphorylation, upregulation of PUMA (apoptosis inducer), and downregulation of survivin (inhibitor of apoptosis), all of which were abolished by FDP treatment. The fate of the cell is determined by p53 in two directions: survival and death (Montero et al., 2013). Physiological levels of ROS trigger protective pathways, whereas p53 acts as a death signal under cytotoxic oxidative stress. It was reported that most p53-induced apoptosis events are dependent on PUMA, a proapoptotic BCL-2 family protein (Jeffers et al., 2003). The role of p53 in regulating carbohydrate metabolism was suggested by inducing its downstream gene, TIGAR (Green & Chipuk, 2006). Under oxidative stress, p53-induced TIGAR expression modulates the glycolytic pathway by lowering the levels of fructose 2,6-bisphosphate, inhibiting glycolysis, and activating the PPP to protect against ROS-induced death (Bensaad et al., 2006). However, under severe IRI, continued TIGAR expression could lead to glycolysis shutdown and inhibition of ATP production with or without PPP activation (Bensaad, Cheung & Vousden, 2009). A TIGAR-null mutation in mice was found to aggravate brain ischemic injury and was associated with low NADPH and GSH levels (Li et al., 2014). In the heart, p53 and TIGAR were also found to enhance hypoxia-induced inhibition of glycolysis and myocyte apoptosis (Kimata et al., 2010). In mild renal IRI, p53-induced TIGAR expression resulted in the inhibition of PFK1 and increased G6PD and NADPH levels, indicating the redirection of the glycolytic pathway to the PPP (Kim, Devalaraja-Narashimha & Padanilam, 2015). However, under severe renal IRI, sustained TIGAR expression did not activate the PPP, and p53 overexpression led to cell apoptosis (Kim, Devalaraja-Narashimha & Padanilam, 2015). The results from this study suggest that tIRI-induced oxidative stress is toxic to the testicular microenvironment, prompting apoptotic pathway activation through p53 overexpression and phosphorylation and preventing additional global tissue damage. The inhibition of glycolysis without activation of the PPP suggests that during tIRI, sustained TIGAR overexpression might redirect glucose metabolism from energy production to nucleotide synthesis for DNA repair.

Conclusion

In conclusion, alterations in the mRNA expression and activities of the major GEs appear to contribute to the tIRI-induced damage to spermatogenesis and hamper cellular energetics. The effects of FDP treatment were very beneficial in sustaining the ATP and antioxidant levels that were depleted by tIRI-induced excessive ROS production and glycolysis shutdown. In addition, the activation of p53 and its transcriptional target TIGAR during tIRI was clearly associated with germ cell apoptosis and did not promote their survival. These findings show promise for FDP application as an adjuvant treatment in the setting of TTD.

Supplemental Information

Data S1 Realtime PCR Data

Click here for additional data file.

Data S2 Glycolytic Enzyme Activities

Click here for additional data file.

Data S3 Histological data

Click here for additional data file.

Data S4 Biochemical Assays

Click here for additional data file.

Additional Information and Declarations

Competing Interests

Author Contributions

Animal Ethics

Data Availability

The authors declare there are no competing interests.

May Al-Maghrebi conceived and designed the experiments, performed the experiments, analyzed the data, contributed reagents/materials/analysis tools, wrote the paper, prepared figures and/or tables, reviewed drafts of the paper, awarded the grant that supported this study: MB 02/15.

Waleed M. Renno performed the experiments, analyzed the data, prepared figures and/or tables, reviewed drafts of the paper.

The following information was supplied relating to ethical approvals (i.e., approving body and any reference numbers):

The animal experimental protocol and procedures were in compliance with the guidelines of the ethics committee on animal research at Kuwait University. The animal study approval number is MB 02/15. The ethical use of animals at Kuwait University is in accordance with the norms of the International Council for Laboratory Animal Sciences (ICLAS).

The following information was supplied regarding data availability:

The raw data has been supplied as Supplemental Dataset.

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
