# Peer review of "Altered expression profile of glycolytic enzymes during testicular ischemia reperfusion injury is associated with the p53/TIGAR pathway: effect of fructose 1,6-diphosphate"

_PeerJ, doi:10.7717/peerj.2195_

## Round 0.1 · original submission · Major Revisions

Dear Dr. Al-Maghrebi,

Thank you for your submission to PeerJ which has now been reviewed by two experts. As you can see, both reviewers have concerns and have provided you with constructive feedback. Please account for their comments in any revision.

Reviewer 1 ·

Basic reporting

Please look below

Experimental design

Please look below

Validity of the findings

Please look below

Additional comments

In the manuscript Altered expression profile of glycolytic enzymes during testicular ischemia reperfusion injury is associated with the p53/TIGAR pathway: Effect of fructose 1,6-bisphosphate” the authors aim to assess the activities and expression of glycolytic enzymes in the testis and their possible modulation during testicular ischemia reperfusion injury (tIRI). The authors also propose to study the role of fructose 1,6-bisphosphate (FBP) on tIRI. The manuscript needs language revision and the references are need update. The Methods are not described in sufficient way to allow reproducibility. Results should be numerically described and discussion needs major revision. Overall the manuscript may be of interest but needs extensive revision before proceed for publication. I have several suggestions that may be useful to improve the manuscript.

a) Language correction is mandatory. There are too many typo and grammar errors. For instance: “Sprague Dawly” instead of “Sprague Dawley”.

b) Introduction concerning the metabolic cooperation between testicular cells should be updated (PMID: 25620223; 23011766; 22549313). The implications of metabolic defects and metabolic diseases to male fertility should also be updated (PMID: 26698679; 26676340; 26064993 among others)

c) The occurrence of Warburg effect in Sertoli cells should also be referred as it highlights the relevance of glycolytic enzymes to spermatogenesis (PMID: 25043918).

d) The selected dosage of 2 g/kg FBP should be better explained

e) Biochemical assays must be better described. The colorimetric assays must have references from Sigma.

f) Please include the primers sequences

g) The description of genes expression should be numeric, more detailed and include statistical analysis. The same is applied for enzymatic activities.

h) Results of testicular oxidative stress must also be numerically described so the reader can clearly evaluate the differences

i) Discussion is very unfocused and needs major revision. The authors present an introduction for each enzyme and its action and slowly introduce their results and conclusion. I would rather prefer to have the other way around. The authors should highlight the main results, briefly introduce the function of the enzymes and put the results in a perspective way. The “take home message” is difficult to understand.

j) I have shortly evaluated the raw data and I have detected some possible outliers using Grubbs’ method, α=0.2. p<0.05. I would recommend the authors to carefully check this issue. Moreover, Pearson correlation coefficients could be performed to determine correlations between the different parameters analysed.

Reviewer 2 ·

Basic reporting

The authors of the manuscript “Altered expression profile of glycolytic enzymes during testicular ischemia reperfusion injury is associated with the p53/TIGAR pathway: Effect of fructose 1,6-bisphosphate” aimed to evaluate the involvement and modulation of glycolytic enzymes during testicular ischemia reperfusion injury. Thy aimed specifically to evaluate the role of fructose 1,6-bisphosphate on that process.
Although the manuscript may potentially present new and interesting results, in its present form it presents multiple weaknesses that should be addressed.
The manuscript needs thorough language revision (grammar and typo errors) and reference update. Some sections need to be more detailed (M&M). The discussion is quite poor and needs a revision in order to address all the major results obtained and to transmit the idea the authors are trying to convey.

Experimental design

The Material and Methods section must be extended in order for the readers to be able to reproduce the experiments. Moreover, the reasoning for the selection of the specific treatment should be better explained. The material used must have the reference of the supplier when relevant.

Validity of the findings

The Results section must include the description of the numeric values so that readers better understand the results obtained and their discussion. A proper statistical analysis of the results obtained should be performed and the results should be discussed in accordance.

Additional comments

This reviewer has several suggestions to improve the scientific impact of the manuscript.
Manuscript - Language editing is mandatory. There are several typo and grammar errors throughout the manuscript.
Introduction - Introduction must be updated. The authors should reference relevant recent publication addressing testicular metabolism and the metabolic cooperation established between testicular cells in physiological and non-physiological conditions (PMID: 23348098; 25620223; 22549313; 26698679; 26676340; 26064993, among others)
Material and Methods – This section must be extended in order for the readers to be able to reproduce the experiments. Moreover, the reasoning for the selection of the FBP dosage (2 g/kg) should be better explained. The material used must have the reference of the supplier when relevant (e.g. antibodies and assay kits)
Results – The results section must include the description of the numeric values and not only the statement of “increase” vs. “decrease”. A proper statistical analysis of the results obtained should be performed (e.g. enzymatic activities, gene expression).
Discussion - this section needs a major revision for it is not perceivable the message the authors are trying to convey to the readers. The “story” behind the results obtained is quite difficult to understand because of the manner this section is constructed.

---

## Round 0.2 · Minor Revisions

At the request of PeerJ I was asked to step in and provide this revision decision. I have considered the prior revision decision and the latest reviewer reports.

Please execute the following minor revisions before we can officially accept the manuscript:

1) Page 5 line 81: Please revise the reference Minutoli et al (2012).
The author list corresponds to PMID 21828180 and does not match the rest of the citation , which belongs to PMID 22549313.

2) Page 6 line 97: Please revise this citation (Oliveira et al, 2015), i.e. correct the typo in the text from 2014 to 2015.

3) There are two citations for Alves et al., 2016. These must be identified separately (such as 2016a and 2016b, respectively) in the text.

4) In the abstract, define the abbreviations GSH and SOD.

We look forward to receiving your revision as soon as possible.

Reviewer 1 ·

Basic reporting

The authors have done a nice work to reply to reviewer's comments.

Experimental design

The authors have done a nice work to reply to reviewer's comments.

Validity of the findings

The authors have done a nice work to reply to reviewer's comments.

Additional comments

The authors have done a nice work to reply to reviewer's comments. Congratulations.

Reviewer 2 ·

Basic reporting

In the manuscript (2016:02:9260:0:1: REVIEW) entitled “Expression profile of glycolytic enzymes during testicular ischemia reperfusion injury is associated with the p53/TIGAR pathway: Effect of fructose 1,6-bisphosphate”, the authors took in consideration all suggestions made by reviewers.

In this form the authors carefully revised the introductory section and added new information regarding testicular metabolism. This section is now significantly improved and gave a clear message. Materials and methods were carefully revised and results were also changed accordingly. The discussion of the manuscript was significantly improved and the authors highlighted the obtained results and put in a perspective way. The message is now more perceivable to the readers.

Experimental design

None

Validity of the findings

None

Additional comments

Minor points should be considered to revision:

Pag. 5 line 81: Please revise the reference Minutoli et al (2012). This work was not published in NRU.
Pag. 6 line 92: The authors should reference relevant recent publication addressing testicular metabolism in non-physiological conditions (PMID: 26148570).
Pag. 6 line 97: Please revise this citation (Oliveira et al, 2015)
Pag. 17 line 361: Relevant publication should be added (PMID: 25346452).
Pag. 17 line 363: The authors should reference relevant work addressing testicular metabolism and sperm quality in diet-treated animals (PMID: 23495257).

---

## Round 0.3 · accepted · Accept

Thank you for the attending to all the editorial and reviewer comments.